

# Are children and dogs best friends? A scoping review to explore the positive and negative effects of child-dog interactions

Claire S. E. Giraudet[1,2], Kai Liu[1,3], Alan G. McElligott[1,2] and Mia Cobb[4]

[1] Department of Infectious Diseases and Public Health, Jockey Club College of Veterinary Medicine and Life Sciences, City University of Hong Kong, Hong Kong SAR, China
[2] Centre for Animal Health and Welfare, Jockey Club College of Veterinary Medicine and Life Sciences, City University of Hong Kong, Hong Kong SAR, China
[3] Animal Health Research Centre, Chengdu Research Institute, City University of Hong Kong, Chengdu, China
[4] Animal Welfare Science Centre, Faculty of Veterinary and Agricultural Sciences, University of Melbourne, Parkville, Victoria, Australia

Corresponding authors
Claire S. E. Giraudet,
claire.giraudet98@gmail.com
Alan G. McElligott,
alan.mcelligott@cityu.edu.hk

## ABSTRACT

Personal wellbeing is greatly influenced by our childhood and adolescence, and the relationships formed during those phases of our development. The human-dog bond represents a significant relationship that started thousands of years ago. There is a higher prevalence of dog ownership around the world, especially in households including children. This has resulted in a growing number of researchers studying our interactions with dogs and an expanding evidence base from the exploration of child-dog interactions. We review the potential effects of child-dog interactions on the physical, mental, and social wellbeing of both species. A search of the SCOPUS database identified documents published between January 1980 and April 2022. Filtering for key inclusion criteria, duplicate removals, and inspecting the references of these documents for additional sources, we reviewed a total of 393 documents, 88% of which were scientific articles. We were able to define the numerous ways in which children and dogs interact, be it neutral (*e.g.*, sharing a common area), positive (*e.g.*, petting), or negative (*e.g.*, biting). Then, we found evidence for an association between childhood interaction with dogs and an array of benefits such as increased physical activities, a reduction of stress, and the development of empathy. Nonetheless, several detrimental outcomes have also been identified for both humans and dogs. Children are the most at-risk population regarding dog bites and dog-borne zoonoses, which may lead to injuries/illness, a subsequent fear of dogs, or even death. Moreover, pet bereavement is generally inevitable when living with a canine companion and should not be trivialized. With a canine focus, children sometimes take part in caretaking behaviors toward them, such as feeding or going for walks. These represent opportunities for dogs to relieve themselves outside, but also to exercise and socialize. By contrast, a lack of physical activity can lead to the onset of obesity in both dogs and children. Dogs may present greater levels of stress when in the presence of children. Finally, the welfare of assistance, therapy, and free-roaming dogs who may interact with children remains underexplored. Overall, it appears that the benefits of child-dog interactions outweigh the risks for children but not for dogs; determination of the effects on both species, positive as well as negative, still requires further development. We call for longitudinal studies and cross-cultural

research in the future to better understand the impact of child-dog interactions. Our review is important for people in and outside of the scientific community, to pediatricians, veterinarians, and current or future dog owners seeking to extend their knowledge, and to inform future research of scientists studying dogs and human-animal interactions.

# INTRODUCTION

The distribution of domestic dogs across almost every ecological niche of our planet has been attributed to their ability to directly interact with humans (*Miklósi & Topál, 2013*). It has been millennia since dogs were domesticated and began interacting with humans (*Zhang, Khederzadeh & Li, 2020*). Dogs (*Canis familiaris*) descended from the ancestral gray wolf (*Canis lupus*) tens of thousands of years ago, making canines one of the first domesticated taxa (*Ostrander et al., 2017*). Natural and artificial selection processes have shaped dogs into what we know today as "man's best friend". It has been suggested that various features of the species including but not limited to morphology, behavior, and cognition have emerged specifically as adaptations to living in social groups with people and to aid communication with humans (*Range & Virányi, 2014*).

The global domestic dog population is estimated to be 900 million (*Gompper, 2013*), of which 20–30% are considered companion animals, the rest being free-ranging individuals (*Hughes & Macdonald, 2013*). Culture and context can define if a dog is considered a companion animal, a divinity, pest, or food (*Jackman & Rowan, 2007*; *Gray & Young, 2011*). Companion dogs, or pets, live in or alongside homes, have a given name, and are very often considered as family members (*Jackman & Rowan, 2007*). Such dog 'ownership' is common, with 38% of American, 40% of Australian, 33% of English, and 21% of French households reporting to care for at least one dog (*American Veterinary Medical Association, 2018*; *Fédération des Fabricants d'Aliments pour Chiens, Chats, Oiseaux et autres animaux familiers, 2018*; *Animal Medicine Australia, 2019*; *Pet Food Manufacturing Association, 2021*). Our understanding of companion dogs practices throughout Asian countries is growing, with China, India, South Korea, and Japan reporting 25%, 5%, 20%, and 17% of dog ownership, respectively (*Growth from Knowledge, 2016*; *Minatoya et al., 2019*; *Cherian, Dugg & Khan, 2020*). People also live alongside the many free-roaming dogs, unrestricted animals commonly found in urban and regional areas of Brazil, China, and India for example (*Kwok et al., 2016*; *Tian et al., 2018*; *Corfmat et al., 2022*). With so many people living alongside domestic dogs, scientific investigations of the effects of human-animal interactions relating to dogs are growing.

Child and adolescent development are fundamentally affected by the relationships they form with others (*Osher et al., 2020*), and the relationship that links humans to dogs is significant. Interacting with animal companions, or pets, has become a normal part of

growing up for many children (*Melson & Fine, 2015*). Families may include a dog before the arrival of a baby in the home, or families including a child may acquire a canine companion during the infant's childhood. The likelihood of owning a dog increases in households with children (*Downes, Canty & More, 2009*; *Holland, 2019*). Moreover, the age of children appears as an important factor affecting dog ownership. Having a companion dog is reported more in families with children between 6 and 10 years old while dogs are less likely to be owned by families with children in other age groups (*Westgarth et al., 2007*; *Murray et al., 2010*). Limited information is known from other parts of the world where less research has been undertaken. For example, in Seoul, South Korea, dogs are mostly owned by single, educated, high-income men, possibly due to a lower proportion of families with children in the studied sample population than in research from comparable countries such as Great Britain (*Westgarth et al., 2007*; *Kim et al., 2020*). If not residing with them, children may encounter dogs in the homes of extended family or friends, or school settings as animals are increasingly involved in education (*Gee, Griffin & McCardle, 2017*). Furthermore, assistance and therapy dogs are receiving wider recognition globally, for helping in roles such as alerting epileptic seizures or visiting children's classrooms to benefit targeted learning outcomes (*Brelsford et al., 2017*; *Correale et al., 2017*; *Catala et al., 2018*). Moreover, the COVID-19 pandemic impacts in 2020-2021 resulted in much greater dog adoption rates and the strict lockdowns in many countries, amplifying time shared with dogs as people transitioned to working and learning from home for extended periods (*Morgan et al., 2020*; *Christley et al., 2021*). This demonstrates there are many settings in which a co-habiting child and a dog might interact. Child-dog interaction studies represent a broad area under investigation with research being conducted with varying aims, methodologies, and measures. This has resulted in sometimes contradictory results between studies, reflecting a diversity of human-dog interactions, classed as beneficial, neutral or non-existent, and sometimes harmful (*Herzog, 2011*; *Friedman & Krause-Parello, 2018*; *Wells, 2019*). Additionally, few studies have addressed the impact of such interactions with people to dogs (*Hall, Finka & Mills, 2019*; *Glenk & Foltin, 2021*).

The purpose of this review is to present an overview of the scientific research on child-dog interactions. We acknowledge that reviews of the effects of child-dog interactions already exist, but these were focused on one type of interaction in particular (*e.g.*, *Purewal et al., 2017*; *Patterson et al., 2022*). Herein, we aim to compile the results of various studies on both positive and negative interactions for the two species to consider the possible effects of such interactions on multi-species' quality of life. To do so, we take into account the three domains (*i.e.*, physical, mental, and social) which are considered as fundamental dimensions of the quality of life and animal welfare (*World Health Organization, 1995*; *Mellor et al., 2020*). We have opted for a scoping review approach, which target broad questions (*Munn et al., 2018*). The specific goals are to (a) describe the variety of child-dog interactions, and (b) summarize the reported outcomes of such interactions on both species. Finally, we close the review by pointing out existing gaps and providing some recommendations for future research to help advance our understanding of child-dog interactions. The results of this scoping review are likely to benefit people in

**Table 1 Full electronic search strategy and identification of additional sources used.**

| Query strings | Sources found | Sources included after key criteria screening | Sources after duplicate removal | Additional sources retrieved from the references |
|---|---|---|---|---|
| KEY (dog-child OR child-dog) OR TITLE (dog-child OR child-dog) | 36 | 23 | 317 | 76 |
| KEY (dog* AND child* OR adolescen*) AND TITLE (dog* AND child* OR adolescen*) | 451 | 280 | | |
| KEY (dog* AND education*) AND TITLE (dog* AND education*) | 41 | 17 | | |
| KEY (dog* AND bite* AND child* OR adolescen*) AND TITLE (dog* AND bite* AND child* OR adolescen*) | 136 | 87 | | |
| KEY (dog* AND welfare AND child* OR adolescen*) OR TITLE (dog* AND welfare AND child* OR adolescen*) | 113 | 32 | | |
| KEY (dog* AND play* AND child* OR adolescen*) OR TITLE (dog* AND play* AND child* OR adolescen*) | 56 | 24 | | |
| Total | 833 | 463 | | 393 |

and outside of the scientific community. It is relevant to pediatricians, psychologists, and veterinarians working with dog-owning people and aspirant dog owners. It can also be used by scientists studying dogs and human-animal interactions to plan future research. Moreover, it is an opportunity for current and prospective dog owners to extend their knowledge of the various outcomes possibly derived from child-dog interactions, hopefully encouraging future positive interactions.

## METHODS

### Protocol

The PRISMA Guidelines for Scoping Reviews were followed to perform this scoping review (*Tricco et al., 2018*). To identify potentially relevant documents, the SCOPUS electronic database was searched. The keywords used to conduct the search were: 'Dog*', 'Child*', 'Adolescen*', 'Child-dog', 'Dog-child', 'Education*', 'Bite*', 'Welfare', and 'Play*' (Table 1). The following inclusion criteria were used to select suitable documents, including articles, book chapters, conference papers and reviews: (a) publication after 1980, ranging from January 1980 to April 2022, (b) publication in English language, (c) focus on the effect of dogs on children (aged ≤17 years) or conversely, the effect of children on dogs. Physical, mental, and social effects were considered where described. Pet dogs, assistance and therapy dogs as well as free-roaming individuals were included. Information was extracted from each included study to achieve the aim of this scoping review. To achieve the first aim, *i.e.*, describe the kinds of child-dog interactions, data items included child characteristics, dog characteristics, behaviors involved, and contexts. To achieve the second aim, *i.e.*, summarize study outcomes, data items included the results of each study.

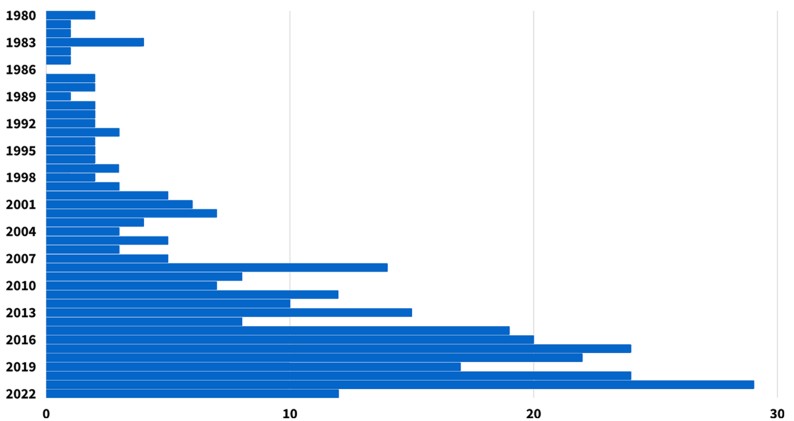

**Figure 1 The number of sources included in this scoping review, presented by their year of publication, from January 1980 to April 2022.**

## Synthesis of results

Our preliminary search resulted in 833 documents (Table 1). We screened results for relevance by reading abstracts and applying the criteria detailed in the protocol above. A total of 317 documents were identified after checking for key criteria and removing duplicates (see Appendix). We then inspected the references of these papers for supplemental studies not yielded by the search, identifying an additional 76 sources. In the end, a diverse sample of 393 documents was selected for the final review, which included studies with a variety of designs, participant ages, types of intervention, and outcome measures. We noticed that overall, there is an increase in the number of publications on the subject from year to year since the beginning of the 21st century (Fig. 1), demonstrating the growing interest in this field of study.

Among the 317 documents identified on SCOPUS, 88% were scientific articles, 10% were reviews, and the remaining 2% were either book chapters or conference papers (Fig. 2). Moreover, we found that most of the scientific articles originated from either Europe (37%) or North America (37%), followed by Asia (12%) and Australasia-Oceania (8%). Regions of the world like South America, Africa, and the Middle East each only accounted for 2% of the total identified articles (Fig. 3).

While recognizing that our SCOPUS search did not provide an exhaustive list of all the documents published on the subject between 1980 and 2022, it nonetheless yielded an evidence base from which to analyze the state of the field. We have opted to focus on a descriptive and qualitative synthesis of the results rather than a meta-analysis, especially considering the heterogeneity across studies.

## HOW CHILDREN AND DOGS INTERACT WITH EACH OTHER

At home, children and adolescents talk to their dogs, share secrets with them and seek comfort from them when sad (*McNicholas & Collis, 2001*; *Kurdek, 2008*; *Hawkins, Williams & Scottish Society for the Prevention of Cruelty to Animals, 2017*; *Hull, Guarneri-White & Jensen-Campbell, 2022*). They may partake in caretaking behaviors toward their dog, as a formal responsibility or by choice (*Muldoon, Williams & Lawrence,*

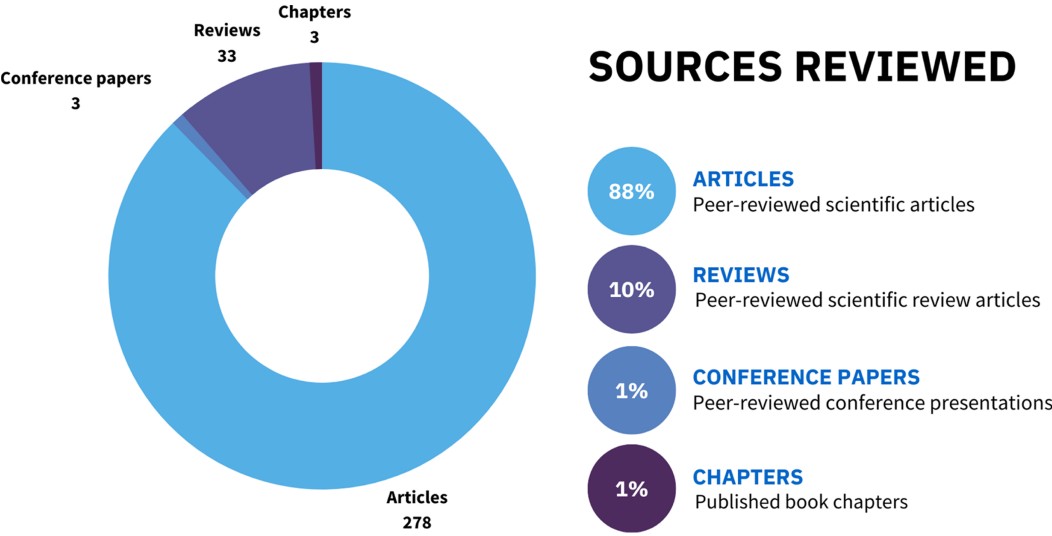

**Figure 2** **The proportional representation of different sources of evidence included in this scoping review.**

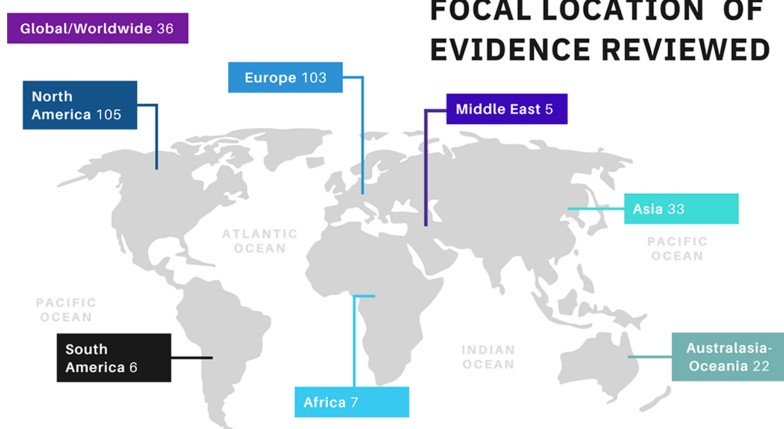

**Figure 3** **The focal locations of different sources of evidence represented in this scoping review.**

*2015*; *Hall et al., 2016*; *Kerry-Moran & Barker, 2018*). Such behaviors include but are not limited to feeding, giving water, and grooming. The impacts of co-sleeping with pets (*i.e.*, action of sharing a bed/bedroom) has been investigated and no differences have been found regarding sleep dimensions (*e.g.*, quality, duration) between children who co-sleep with their dogs and those who do not (*Rosano et al., 2021*; *Rowe et al., 2021*). Dogs need to be taken outside for different reasons, such as toileting, exercise, and to socialize with people or other dogs. Children and adolescents can take part in dog walking with or without the rest of their family (*Wenden et al., 2021*; *Coci, Saunders & Christian, 2022*; *Christian et al., 2022*). Additionally, dog training can play a major role in the good functioning of human and dog cohabitation and children sometimes participate. Children and dogs can also be excellent playmates (*Boisvert & Harrell, 2021*). However, aggressive

interactions can be witnessed between a child and a dog. Dog bites are common injuries treated in pediatric departments of hospitals all over the world (*d'Angelo et al., 2022*; *Patterson et al., 2022*). Children cruelty toward animal also exists and should be taken very seriously (*McDonald et al., 2018*; *Hawkins, Scottish Society for the Prevention of Cruelty to Animals & Williams, 2020*), especially in light of the literature linking animal cruelty and other forms of violence (*e.g.*, *Gullone & Robertson, 2008*).

The purpose of animal-assisted interventions is to enhance human quality of life with animals as therapeutic adjuncts. Due to their ease of training and availability, dogs are the most commonly used animals (*Glenk & Foltin, 2021*). Animal-assisted education has become more and more common in numerous nations, particularly in Australia and the United States (*Gee, Griffin & McCardle, 2017*; *Grové et al., 2021*), which has given rise to studies investigating the impact of "the presence of a dog in the classroom" (*Gee, Fine & Schuck, 2015*; *Brelsford et al., 2017*). Animal-assisted education can take the form of dogs actually assisting during children education, for example when a child has the opportunity to read to a dog (*Hall, Gee & Mills, 2016*; *Lenihan et al., 2016*; *Noble & Holt, 2018*; *Henderson et al., 2020*). Japanese programs are different because they take the form of "education through assisting animals", meaning that pupils are taught animal-rearing (*Nakajima, 2017*). In medical settings, it is sometimes possible to interact with a dog prior to, during, or after a medical exam as part of programs where dogs regularly visit hospitals and clinics (*Chur-Hansen et al., 2014*; *Vagnoli et al., 2015*; *Vincent, Heima & Farkas, 2020*). In those studies, dogs are not just present as static ornaments in the room. Children are encouraged to interact with the animals, here classified as therapy-dogs, by petting them or engaging with them, for example, by asking them to perform a trick in exchange of a food reward (*Gee, Harris & Johnson, 2007*; *Gee, Crist & Carr, 2010*; *Gee et al., 2012*). Dogs classified as assistance-dogs are trained to assist people in their daily lives. Nowadays, such dogs can be found assisting people with epilepsy to know when seizures are imminent (*Catala et al., 2018*) and children on the autistic spectrum to reduce symptom severity and repetitive behaviors, as well as improve motor skills and communication (*Wright et al., 2015*; *O'Haire, 2017*; *Ben-Itzchak & Zachor, 2021*; *Nieforth, Schwichtenberg & O'Haire, 2021*).

The stray or free-roaming dogs that can be found in many countries are those that live unrestricted lives and do not depend on any specific human for food (*Jackman & Rowan, 2007*; *Rahaman, 2017*). Partly due to the difficulty in finding a universal definition, there is considerable variation between countries regarding the percentage of stray dogs. These dogs are often blamed for disease spread and attacks, which may explain why they are sometimes harassed or killed inhumanely using poison (*Cleaveland et al., 2006*; *Jackman & Rowan, 2007*).

In this section, we outline many of the ways and settings in which dogs and children can interact. To better understand the importance of these interactions to both children and dogs, we describe the valence of the interactions, considering physical, mental, and social effects.

# BENEFICIAL EFFECTS OF CHILD-DOG INTERACTIONS FOR CHILDREN

## Physical benefits

Regular physical activity is vital for the development and growth of children (*World Health Organization, 2019*), with strong evidence supporting numerous health benefits for children and adolescents (*Poitras et al., 2016*). Conversely, sedentary behaviors have been identified as a major public health concern linked to poor health outcomes (*Carson et al., 2016*). Strict lockdowns and similar COVID-19-related policies in 2020–2021 are believed to have exacerbated sedentary behaviors because of the transition to working and learning from home and advice to "stay home" or indoors (*Bates et al., 2020*). For children and adolescents, living with a dog increases the likelihood of getting the recommended level of weekly physical activity either through dog walking or through active play with a dog (*Christian et al., 2013*; *Engelberg et al., 2015*; *Martin et al., 2015*). Several studies have argued that the amount of physical activity in those children is strongly associated with their level of attachment to their dogs, with children displaying stronger attachment to their dog being more likely to walk them (*Westgarth et al., 2013*; *Gadomski et al., 2017*; *Linder et al., 2017*).

Walking is one of the many activities people and dogs can share. Caution should always be exercised when giving the leash to a child. Apart from teaching children to attend to dog signaling, which we will discuss later in this review (see "Dog Bites"), parents must make sure that their child possesses the physical strength necessary to avoid being pulled or dropping the leash to prevent dangerous situations. Children that live with a dog spend more time walking and are more physically active (*Salmon et al., 2010*; *Christian et al., 2013*; *Martin et al., 2015*). Children who walk their dog are more likely to be independently mobile (*i.e.*, walking or cycling without adult supervision) than those who do not (*Christian et al., 2014*, *2022*). Moreover, a pilot study on therapy dog-walking for adolescents with orthopedic limitations yielded promising results with an increase of physical activity throughout the duration of the program, a high attendance rate, and a perceived positive experience by the adolescents involved (*Vitztum, Kelly & Cheng, 2016*). However, not all children living with a dog partake in dog walking. In Australia, 41% to 45% of children who live with a dog do not actually walk it (*Salmon et al., 2010*; *Christian et al., 2014*) while 43% of English dog-household children are reported to not participate in routine dog walking (*Westgarth et al., 2013*). Hence, even though children who live with a dog seem to present higher levels of physical activity on average (*Engelberg et al., 2015*; *Christian et al., 2022*), it is not the case for every child and may be through activities other than dog walking.

Playing is an activity that children value and need and children believe that dogs share their love of play (*Muldoon, Williams & Lawrence, 2015*; *Boisvert & Harrell, 2021*). Children consider their dogs as special friends and playmates (*Melson, 1990*; *Muldoon, Williams & Lawrence, 2015*; *Muldoon et al., 2019*). They enjoy engaging in many sorts of play with their companions, for example running around with the animal and engaging in fetch activities (*Boisvert & Harrell, 2021*). Living with a dog also increases children's time

spent in outdoor play which is associated with higher levels of independent mobility (*Christian et al., 2014*, *2022*). However, the prevalence of pet play appears negatively related to age. Indeed, children in primary school are more likely to play with their dog compared to children in secondary school (*Martin et al., 2015*). This could be due to the general trend of physical activity decline during adolescence (*Dumith et al., 2011*). While playing is an enjoyable activity for both children and dogs, it can also lead to accidents and injuries. Hence, experts recommend that dog-child interactions should always be supervised by adults so as to mitigate risks and avoid mishaps (*Rezac, Rezac & Slama, 2015*; *Jakeman et al., 2020*).

Given children who live with a dog demonstrate a higher likelihood of achieving the recommended levels of physical activity (*Christian et al., 2013*), it is worth considering if dogs may impact childhood obesity. Dogs have the potential to motivate obese children for physical activity thanks to the provision of a surrogate support network; this could trigger implicit motives which enhance motivation for activity (*Wohlfarth et al., 2013*; *Linder et al., 2017*). Although other studies agree on the potentially beneficial effect of dogs, results regarding dog interactions and children's weight status are inconsistent, likely reflecting the complexity of factors which relate to childhood obesity (*Timperio et al., 2008*; *Westgarth et al., 2012*, *2017*; *Christian et al., 2013*; *Gadomski et al., 2017*). One explanation could also be that interacting with a dog might not be vigorous or sustained enough as physical activity to significantly affect childhood weight status (*Westgarth et al., 2017*).

During potentially stressful tasks such as medical examinations, knowledge tests, or interviews, the presence of a friendly dog (familiar or not) can impact human physiology by helping lower children's heart rate and blood pressure (*Friedmann et al., 1983*; *Vormbrock & Grossberg, 1988*; *Nagengast et al., 1997*; *Krause-Parello et al., 2018*); although not all studies agree with that conclusion (*Grossberg, Alf & Vormbrock, 1988*; *Schretzmayer, Kotrschal & Beetz, 2017*; *Kerns et al., 2018*). Such results may be derived from and/or enhanced by physical contact such as petting a dog, which might affect sympathetic arousal (*Beetz et al., 2011*).

Research on the association between dog exposure and the incidence of allergies in children has provided mixed results (*Apfelbacher et al., 2016*). Some findings suggest that living with a dog may increase exposure to endotoxins, which can enhance children's immune systems and reduce the likelihood of becoming sensitized, therefore protecting them from the development of allergies (*Campo et al., 2006*; *Lødrup Carlsen et al., 2012*). *Epstein et al. (2011)* concluded that living with a dog significantly reduced the likelihood of eczema in young children. Conversely, other research concluded that living with a dog was associated with higher incidence of pet allergy and asthma in children (*Collin et al., 2015*; *Pyrhönen, Näyhä & Läärä, 2015*; *Luo et al., 2018*; *Mendy et al., 2018*). These results will be discussed later in this review (see "Zoonotic Infections, Asthma, and Allergies").

## Mental and social benefits

For children and adolescents, living with a dog has been associated with a decreased likelihood of anxiety and stress (*Covert et al., 1985*; *Gadomski et al., 2015*). Children aged 7–8 years see dogs as useful protectors and supporters in scary situations (*McNicholas &*

*Collis, 2001*). As children grow, their use of parental support for stress coping diminishes and they partially replace it with other social support figures, which may include pets (*Kertes et al., 2017*). Dogs seem to be able to provide a "social buffer" *via* neurochemical responses, in particular by reducing the levels of plasma cortisol (*Beetz et al., 2012*), thus reducing children and adolescents anxiety, specifically separation and social anxiety (*Gadomski et al., 2015*; *Wright et al., 2015*). Adolescents living with a dog reported lower feelings of loneliness, a precursor for anxiety, depression and low self-esteem (*Black, 2012*). Scientists argue that the support and benefits in stress reduction derived from dogs is strongly correlated with the time spent in physical contact with them: the more time spent stroking a dog, the more apparent and longer-lasting the benefits (*Beetz et al., 2012*).

Consistent with studies examining human adults, dogs facilitate social interactions between children (*Mader, Hart & Bergin, 1989*; *Christian et al., 2020*). Children interacting with dogs tend to have increased confidence and decreased fear of social rejection (*Purewal et al., 2017*). Child-dog interactions also reduce the incidence of aggressive behaviors from at-risk students and increase positive social behaviors and empathy (*Gee, Griffin & McCardle, 2017*). Thereby, children and students living with dogs (or with pets in general) can be considered as better socially integrated, with wider social networks, and more popular with their classmates (*Beetz et al., 2012*). Dogs may protect children from developing peer relationship problems, emotional symptoms, and deficits in prosocial behaviors (*e.g.*, sharing, helping, and cooperating) (*Christian et al., 2020*). It may be that interacting with dogs offers opportunities for children to learn about social concepts through mimicking the interactions that they would have with other humans (*Christian et al., 2020*). Additionally, the "stay-at-home" orders as well as the social distancing measures that came with the COVID-19 restrictions have brought challenges to many. Studies from all over the world agree that during the 2020–2021 lockdowns, dogs helped prevent loneliness in adults and children (*Morgan et al., 2020*; *Young et al., 2020*; *Bussolari et al., 2021*; *Martin et al., 2021*; *Oliva & Johnston, 2021*; *Lee, Song & Lee, 2022*). Before the pandemic, the positive effect of dogs against loneliness for children was already being highlighted (*Rew, 2000*; *Black, 2012*; *Purewal et al., 2017*).

Children who live with a dog may display better development of empathy and emotional wellbeing (*Vidović, Štetić & Bratko, 1999*; *Svensson, 2014*). Conveying one's emotions is a key aspect of communication, and the ownership of pets in toddlerhood may promote a child's ability to express their emotions (*Sato et al., 2019*). Because dogs are seen as non-judgmental partners to whom they can confide in (*Kerns et al., 2018*; *Gee et al., 2021*), interacting with a dog can show to children that they are allowed to express their emotions. Child-dog interactions in the school setting have been linked to reduced aggression in at-risk students and increased positive social behaviors and empathy (*Hergovich et al., 2002*; *Gee, Griffin & McCardle, 2017*). Children as young as 7 years old who share their home with dogs can demonstrate extensive knowledge about canine needs, although they sometimes lack the confidence to share their knowledge (*Muldoon, Williams & Lawrence, 2016*). Interestingly, an emotional connection seems to be a necessary prerequisite to the recognition of needs (*Muldoon, Williams & Lawrence, 2016*). Moreover, children from these households were significantly less accepting of animal cruelty

(*Hawkins, Scottish Society for the Prevention of Cruelty to Animals & Williams, 2020*). These studies show that dogs can be important to the development of empathy in children, also relating to childhood and adolescents' regulation and expression of emotions and appropriate behaviors in social settings, such as schools. Empathy is recognized as an important component trait underlying our duty of care, or responsibility, toward other people and animals (*Glanville, Hemsworth & Coleman, 2020*).

Taking care of a dog, taking it for walks, feeding it, and playing with it can promote children certain social values and skills, for example a sense of responsibility for the welfare of others (*Muldoon, Williams & Lawrence, 2015*, *2016*). After completing a series of structured discussions with children aged seven to 13 years old, *Muldoon, Williams & Lawrence (2015)* concluded that despite the fact that there is wide variation in the degree to which children look after family companions, direct interaction with pet animals, such as through play, may allow children to develop a natural sense of care for them. They also showed that children with the most responsibility when it came to their pets presented more extensive knowledge regarding their needs and welfare. This is in tune with parents who often report acquiring a dog with the goal of teaching their children responsibility (*Melson & Fine, 2015*; *Jalongo & Ross, 2018*); the benefits of caring for animals are viewed positively by parents (*Covert et al., 1985*). We believe it important to clarify the difference between "caring about" and "caring for". Despite the fact that dogs are among children's favorite animals (*Borgi & Cirulli, 2015*), indicating that many children genuinely care about them, it does not always result in direct caretaking behaviors. For example, pet care is not routinely performed by pre-adolescents (*Davis, 1987*). For different reasons, children may not specifically interact with their dogs apart from playing with them and joining in on family walks. It may be by choice from the child, or because they rely on their parents (most often mothers) to take care of their animal companions (*Davis, 1987*; *Muldoon, Williams & Lawrence, 2015*). However, some children report that their parents will not allow them to directly care for the family dog or interact with them in ways they would like (*Muldoon, Williams & Lawrence, 2015*, *2016*). *Covert et al. (1985)* showed that adolescents who took care of animals including dogs reported gaining responsibility but their study did not examine the degree of involvement in caretaking behaviors. Moreover, gender may play a role in task distribution with girls more often assuming the role of the caregiver, for example (*Muldoon, Williams & Lawrence, 2015*). Research on the effects of gender in child-dog interactions should be carried out in more countries to consider if these findings can be generalized. Nonetheless, when possible, interacting directly to care for dogs may facilitate the acquisition of certain habits that could contribute greatly to children's life skills both at home and at school such as autonomy, self-reliance, and empathy (*Vidović, Štetić & Bratko, 1999*; *Muldoon, Williams & Lawrence, 2016*).

Dogs can be beneficial to educational outcomes for young people. For children 4 to 5 years of age, living with a dog may aid to facilitate their learning and development (*Svensson, 2014*). Children believe that animals, especially dogs, give them their full attention which can increase their sense of importance, satisfaction in learning, and their motivation to learn more (*Svensson, 2014*). Living with dogs during childhood may diminish the risk of developmental delay in the communication and gross motor domains

(*Minatoya et al., 2019*). In schools, dogs create an enjoyable atmosphere which has the potential to improve children's adherence to instructions and to affect pupils engagement, motivation and self-efficacy (*Beetz, 2013*; *Gee, Griffin & McCardle, 2017*). Weekly visits from dogs can improve classroom attitudes toward school attendance and learning (*Beetz, 2013*). Evidence suggests that dog-assisted reading programs may have a beneficial effect on a number of behavioral processes, all of which can contribute to a positive effect on the environment in which reading is practiced, leading to enhanced reading efficiency (*Hall, Gee & Mills, 2016*). However, a review by *Hall et al. (2016)* pointed out the low quality of the evidence base in many studies looking at the effects of dog-assisted reading programs. The growing attention being given to such programs should enable future studies to yield bigger sample sizes as more children may get access to said programs. The peer-review process is a fundamental aspect of scientific publishing that should be favored.

It appears that interacting with a dog, either at home or at school, is generally beneficial for children and teenagers' development when looking at their cognition, socialization, emotions, behaviors, and education (*Purewal et al., 2017*). Thus, the potential of dogs to influence even one of these aspects could impact all the others, possibly for the best.

# RISK OR DETRIMENTAL EFFECTS OF CHILD-DOG INTERACTIONS FOR CHILDREN

## Dog bites

Dog bite injuries are a world-wide problem and every year, thousands of cases of dog bites in children are recorded (see Table 2). Furthermore, those numbers are most likely not representative of the true extent as many incidents go unreported (*Beck & Jones, 1985*). While the majority of reported accidents involve family or neighboring companion dogs biting children (*Bernardo et al., 2002*; *Park et al., 2019*), free-roaming dogs can also present a threat in some countries (*Georges & Adesiyun, 2008*; *Tenzin et al., 2011*; *Mustiana et al., 2015*).

Dog bite incidents are generally attributed to the ignorance of indicators of early discomfort in dog behavior, such as lip licking or head turning away (*Bradshaw & Rooney, 2016*; *Mariti et al., 2017*). This can lead to dogs escalating their behavior when feeling threatened, consequently increasing the risks of growling and bites (*Owczarczak-Garstecka et al., 2018*). Some dogs may skip early behavior signals of discomfort (for example, if acutely hurt) depending on the situation, context, and history of that individual dog. For this reason, dogs can sometimes appear unpredictable. Dogs have strong jaws and teeth designed for tearing and crushing (*De Munnynck & Van de Voorde, 2002*), making them capable of hurting and even killing people. There is a widespread lack of understanding and knowledge of safety practices for dog-child interactions among owners (*Meints, Brelsford & De Keuster, 2018*), which can contribute to the incidence of children being bitten by dogs. In adults, dog bites most commonly involve the extremities (*e.g.*, hands) (*Overall & Love, 2001*), but because of their smaller size, children are more prone to bites to the head and neck (*Oginni et al., 2002*; *Eppley & Schleich, 2013*; *Cavalcanti et al., 2017*; *Hurst et al., 2020*). Such events can understandably lead to children developing a

**Table 2 Research papers (in alphabetical order according to first author) analyzing dog bites in various countries.**

| Reference | Study type | Country | Years | Sample size of dog bite victims | Victim age | Victim gender | Outcome |
|---|---|---|---|---|---|---|---|
| *Alberghina et al. (2017)* | Retrospective review | Italy | 2012–2015 | 140 | 1–84 years | 57 females 83 males | Dog bite rates were the highest among children aged 0–9 years. In children, most injuries were sustained on the head/neck region, whereas in adults, most bites happened on the hands. |
| *Barrios et al. (2019)* | Retrospective review | Chile | 2009 | 4,579 | 0–65+ years | 1,929 females 2,650 males | Dog bite rates were the highest among children aged 5–9 years. Familiar dogs were responsible for most of the incidents which most likely involved the extremities. |
| *Chiam et al. (2014)* | Retrospective review | Australia | 2009–2011 | 277 | 0–17 years | 117 females 160 males | Dog bite rates were the highest among children aged 0–4 years, and injury incidence declined with age. The vast majority of incidents happened in a familiar environment and involved a familiar dog biting the head/neck region. |
| *Cohen-Manheim et al. (2018)* | Retrospective review | Israel | 2009–2016 | 986 | 0–75+ years | 374 females 612 males | Dog bite rates were the highest among children aged 0–14 years, and injury incidence declined with age. Half of the incidents occurred in the street and a quarter in the home. Almost half of the injuries were sustained on the head/neck region. |
| *McGuire et al. (2018)* | Retrospective review | Canada | 2015–2017 | 158 | 0–16 years | 73 females 85 males | Half of the patients were less than 5 years old. Most injuries were sustained on the face, caused by the family pet, with the dog owner present. |
| *Ogundare et al. (2017)* | Retrospective review | Nigeria | 2010–2014 | 84 | 0–18 years | 24 females 60 males | Dog bite rates were the highest among children aged 6–12 years. The lower limb was the commonest bite site. |
| *Park et al. (2019)* | Retrospective review | South Korea | 2011–2016 | 9,966 | 0–93 years | 5,446 females 4,520 males | There have been increases in the rate of dog-bite injury from 2011 to 2016 for both females and males. Dog bite rates were the highest among children aged 7–12 years. In children, most injuries were sustained on the head/neck region, whereas in adults, most bites happened on the upper extremities. |
| *Ramgopal et al. (2018)* | Retrospective review | USA | 2007–2015 | 14,311 | 0–90 years | 7,735 females 6,576 males | Almost thirty percent of the dog bites occurred in patients less than 18 years old, and injury incidence declined with age. In underage patients, dog bite rates were the highest among children aged 7–12 years. In children, most injuries were sustained on the head/neck region, whereas in adults, most bites happened on the upper extremities. |
| *Tenzin et al. (2011)* | Hospital-based questionnaire survey | Bhutan | 2009–2010 | 324 | 0–80 years | 123 females 201 males | Dog bite rates were the highest among children aged 5–9 years. Stray dogs were responsible for most of the incidents, increasing the chances of rabies infection if not treated in time. |
| *Westgarth, Brooke & Christley (2018)* | Interviews | UK | 2015 | 170 | 5–76+ years | 76 females 94 males | Forty-four percent of adults reported having been bitten by a dog during childhood. More than half of the incidents involved a dog they had never met before. Among the 48 children surveyed in this study, only three had been bitten. |
| *Weyer et al. (2020)* | Retrospective review | South Africa | 2015–2017 | 411 | 0–89 years | 227 females 184 males | A quarter of the patients were less than 10 years old. |

subsequent fear of dogs, life-threatening medical conditions, or psychological consequences like Post-Traumatic Stress Disorder (*Peters et al., 2004*; *Ji et al., 2010*). Sadly, in some cases, severe dog bites can even result in death (*Cataldi, Yamout & Glick, 2011*; *Mora et al., 2018*).

Children, especially toddlers, are capable of unpredictable behaviors and can be prone to risk-taking (*Davis et al., 2012*). Most dog bites happen when a young child is left alone with a dog without adult supervision (*Schalamon et al., 2006*). Boys seem at a higher risk of being bitten than girls (*Schalamon et al., 2006*; *Dwyer, Douglas & van As, 2007*; *Messam et al., 2018*; *Zangari et al., 2021*). The nature of human-dog interactions may differ based on gender and therefore play an etiological role in the differences of dog bite frequency between males and females (*Overall & Love, 2001*). Indeed, gender differences in owner-dog interactions have been highlighted (*e.g.*, verbal communication: *Prato-Previde, Fallani & Valsecchi, 2006*; caring behavior: *Muldoon, Williams & Lawrence, 2015*; *Hawkins, Williams & Scottish Society for the Prevention of Cruelty to Animals, 2017*). Moreover, it has been determined that children younger than five are at the highest risk for severe dog bites and those children are most often bitten in their own home by the family dog (*Bernardo et al., 2002*). This has resulted in strong recommendations for child-dog interactions to always be supervised by adults, or for dogs and young children to be physically separated when necessary as a means to prevent injuries or even death to children (*Messam et al., 2018*; *Meints, Brelsford & De Keuster, 2018*). However, in order for the child-dog relationship to develop, children and dogs should not be separated all the time because of the risks of bites. Rather they should be given the opportunity to interact while under appropriate supervision of an adult.

Biting incidents are influenced by a variety of factors. First, there are risk factors associated with dog characteristics such as a previous history of aggressive behavior, sexually intact males, and purebred dogs (*Shuler et al., 2008*; *Casey et al., 2014*). However, the cause of dog bites is often attributed to humans rather than dogs. The most common reasons for a bite to occur are resource guarding and pain-inducing interactions (*Reisner, Shofer & Nance, 2007*). Up to 86% of accidents at home are triggered by child-initiated interactions such as approaching the dog while eating or surprising it while sleeping (*Kahn, Bauche & Lamoureux, 2003*). Dog bite events can also take place during play sessions, either by accident (*e.g.*, a dog biting the hand of a child holding a toy), or because children are being too rough in their play, which can lead to stress and/or pain in dogs which may then result in a bite (*Messam et al., 2008*; *Hall, Finka & Mills, 2019*). The safety of young children mainly relies on adequate observation through adult supervision, their understanding of dog behavior, and anticipatory guidance of the adults around them (*Meints, Brelsford & De Keuster, 2018*). Yet, it is sometimes parents who demonstrate risky reactions and even encourage their children to interact with dogs despite knowing very little about the animal's safety or disposition (*Morrongiello et al., 2013*). For example, the posing of babies, toddlers, and young children on or inappropriately close to dogs for photographs, all the while possibly overlooking potential signals of stress or discomfort in dogs. Such interactions can thus be perceived as negative by dogs, and can potentially lead them to bite in some cases. Hence, it is crucial for parents to realize that safe cohabitation is

based on mutual understanding of interspecific signaling, social gestures, and responsive interactions.

Dogs have been shown to be good at interpreting human signaling, they are quite sensitive to our attentional state (*Kaminski et al., 2017*). People, on the other hand, do not seem to share the same capacity to read dog visual signaling (*Borgi & Cirulli, 2016*; *Jalongo, 2018*; *Csoltova & Mehinagic, 2020*). Although aggression is generally the most readily recognized expression (*Lakestani, Donaldson & Waran, 2014*), children often misinterpret aggression in the facial expression of dogs (*i.e.*, baring of teeth, *Bradshaw & Rooney, 2016*) as happy and smiling (*Meints, Racca & Hickey, 2010*; *Meints, Brelsford & De Keuster, 2018*) with dangerous consequences. Some adults have also been noted to interpret dog behavior in this way (*Demirbas et al., 2016*). Because of their paedomorphic, or baby-like, features (*Waller et al., 2013*; *Kaminski et al., 2019*), dog facial configurations are often perceived as cute, which may result in humans giving a positive appraisal when interpreting canine behavior (*Borgi et al., 2014*; *Borgi & Cirulli, 2016*).

Children as well as adults regularly do not notice dog stress signaling or misinterpret dog attempts to signal their distress (*Meints, Brelsford & De Keuster, 2018*). What is even more disturbing is that even when children do recognize a fearful dog, many are still inclined to approach it which demonstrates a lack of understanding of how to behave appropriately around dogs (*Aldridge & Rose, 2019*). However, adults are able to recognize and classify dog-barking situations (*Pongrácz et al., 2005*; *Silva et al., 2021*) as well as dog growls (*Faragó et al., 2017*). Children also show capacity to understand basic inner states of dogs when listening to acoustic signals from a young age, with older children able to classify barks with superior accuracy (*Pongrácz et al., 2011*; *Eretová et al., 2020*).

There has been a surge in the number of dog bite cases in children during the COVID-19 pandemic (*e.g.*, reported three-fold increase in an American hospital, *Dixon & Mistry, 2020*; 69% increase in an Italian hospital, *Parente et al., 2021*; 78% increase for boys and 66% increase for girls in a British hospital, *Tulloch et al., 2021*). Because of the "stay-at-home" orders put in place around the world, dog exposure increased for children living with dogs, representing more time together and subsequently more opportunities for dog bites to occur (*Christley et al., 2021*). *Dixon & Mistry (2020)* offered three main contributing factors to this rise in dog bites: (1) increased child-dog exposure, similar to summer months when the highest number of dog bites are reported annually, (2) increased level of dog stress, and (3) decreased level of adult supervision. Additionally, the rise in dog adoption during the pandemic may have also played a role. It is possible that this was the first dog for some families and a lack of knowledge in dog behavior can lead to bite incidents (*Meints, Brelsford & De Keuster, 2018*). Furthermore, in the case of puppies, the government "stay-at-home" orders may have prevented them from being adequately trained and/or socialized, which might have led to the development of unwanted behaviors. All studies, either pre- or during the pandemic, come to the same conclusion: effective public communication to improve understanding of the risk factors for dog bites is required. It is possible to prevent these incidents.

There are two main ways to prevent dog bites, namely educating people or modifying the environment, for example by installing fencing/gate barriers within the home to ensure

physical separation between dogs and children (*Shen et al., 2016*). A sterilization program in India led to a decrease of the number of dog-bite cases, possibly by reducing the maternal protective behavior of street dogs, as well as reducing the total number of roaming dogs (*Reece, Chawla & Hiby, 2013*). Furthermore, different educational interventions have emerged over the years, from books to websites (*Schwebel et al., 2016*; *Jakeman et al., 2020*). Bite prevention programs are being used in many countries, targeting both children and adults, and present promising results with a reduction in the prevalence of dog bites and/or a decrease in injury severity (*Boat, 2019*; *Isparta et al., 2021*; *Kienesberger et al., 2022*).

Different types of education programs designed to decrease the incidence of dog bites exist. With the help of an accredited handler and their dog, introducing primary children for 30 min to the "do's and don'ts" of how to behave around dogs increased precautionary behavior (*Chapman, 2000*). Such workshops also represent good opportunities to teach children responsible dog ownership and canine welfare (*Baatz et al., 2020*). Training children and adults to recognize dog signaling behavior using pictures or videos can increase accuracy in their interpretations (*Wilson, Dwyer & Bennett, 2003*; *Lakestani & Donaldson, 2015*; *Meints, Brelsford & De Keuster, 2018*), while presenting children with testimonials of actual dog-bite experiences from adults increased child safety knowledge and lowered their risk-taking around dogs (*Shen, Pang & Schwebel, 2016*). It is very important to educate adults as well as children, given the high proportion of bites that occur when children are still too young to be taught (*Ogi & Colossi, 2016*; *Fein et al., 2019*). The benefits of teaching people how to understand dogs extend beyond the associated decrease in dog bites; education programs can enhance the probability of future positive child-dog interactions.

## Zoonotic infections, asthma, and allergies

Dogs can be a major reservoir of various zoonotic diseases (*Ghasemzadeh & Namazi, 2015*; *Pathak & Kaphle, 2019*). The numerous ways that humans and dogs interact, be it neutral (*e.g.*, sharing a common area), positive (*e.g.*, petting), or negative (*e.g.*, biting incident), can represent opportunities for diseases transmission between both species. At the beginning of the 21st century, over 60 zoonotic infections transmissible to people by dogs had been identified (*Macpherson, Meslin & Wandeler, 2012*). With the COVID-19 crisis of 2020–2021, public interest in diseases transmissible by animals, including those we live with as companions like dogs, has grown anew, as research on the role of pets in the transmission of the pandemic virus can attest (*Bosco-Lauth et al., 2020*; *Shi et al., 2020*; *Dróżdż et al., 2021*).

The proportion of dogs carrying human pathogens is substantial (*Baxter & Leck, 1984*) and infectious diseases that develop in dogs can have a high zoonotic significance and may transmit to humans (*Pathak & Kaphle, 2019*; *Overgaauw et al., 2020*). One example is the high prevalence of rabies in places such as in Nigeria and Tanzania (*Mshelbwala et al., 2021*; *Sikana et al., 2021*), despite the existence of a vaccine for both people and dogs (*Ghasemzadeh & Namazi, 2015*). To this day, several thousand people die each year (estimated at 59,000 annually, *Hampson et al., 2015*) due to rabies, and up to 99% of these

deaths are attributed to the transmission of the virus through dog-bites (*World Health Organization, 2021*). This disease particularly affects children, especially in poor communities (*World Health Organization, 2013*). In Bangladesh for example, most of the victims are children below 15 years old living in lower socio-economic rural communities (*Hossain et al., 2012*). This is attributed to a lack of access to the vaccine and life-saving treatment (*i.e.*, post-exposure prophylaxis) for economic and/or availability reasons, as well as a lack of knowledge about the disease (*Knobel et al., 2005*). Numerous dog rabies' prevention and control programs exist (*e.g.*, China: *Miao et al., 2021*; India: *Gibson et al., 2022*; Namibia: *Athingo et al., 2020*; Nigeria: *Mshelbwala et al., 2021*; Philippines: *Amparo et al., 2019*), which aim to eradicate the disease.

Fortunately, not all dog-borne zoonoses have the capacity to be lethal. For example, a common tapeworm (*Dipylidium caninum*) of dogs and cats can occasionally be found in humans, especially in children, and causes pruritus in the infected host (*Pathak & Kaphle, 2019*). Dog transmitted infections often go unnoticed (*Macpherson, 2005*). Those diseases can be transmitted by simple contact with the infected dogs (petting, hugging), or by infected urine or feces, saliva, or aerosols (*Pathak & Kaphle, 2019*). Children, especially toddlers, are prone to geophagia (eating soil) and a positive association between this practice and the prevalence of toxocarosis (parasitic disease acquired by ingesting infective eggs) has been found in Polish children (*Kroten et al., 2018*). Consequently, children should be protected against such preventable conditions.

Pet ownership in families with children has also attracted considerable research attention due to its potential relationship in the development of asthma and allergies. Growing up with a pet corresponds to an early-life environmental exposure that may impact the development of respiratory conditions such as asthma and allergies (*Medjo et al., 2013*; *Pyrhönen, Näyhä & Läärä, 2015*; *Fall et al., 2015*; *Mendy et al., 2018*). Living with a companion animal is common within households in countries where the incidence and prevalence of childhood asthma have changed considerably over the past decades (*Collin et al., 2015*). Evidence of a positive association between childhood dog exposure and asthma has been found in several studies (*Collin et al., 2015*; *Alqahtani et al., 2017*; *Luo et al., 2018*). Moreover, we are witnessing an increase in the frequency of allergy to these animals in Global North countries (*Dávila et al., 2018*). Indeed, because of the increasing exposure to animals around the world linked to the growing popularity of pet ownership, more people are being diagnosed with pet-related allergies. Yet, the conclusion that living with a dog is linked with a higher incidence of pet allergy comes with no shock as it is no surprise that people exposed to animals are more likely to trigger an allergy to them. But what of the people who are allergic to pets but simply do not know it? There may be a bias of reporting in non-owners as the absence of exposure potentially inhibits the trigger of the allergy, a bias that should be investigated in the future.

There are eight identified dog allergens, named Can f1 to Can f8, that can be found in dog hair, saliva, and urine (*Li et al., 2021*). Allergies to dogs mainly affect a child's respiratory system, and have been identified as the causal factor for asthma, rhinoconjunctivitis, and atopic dermatitis (*Li et al., 2021*). Identification of pet-related allergies is increasing in China, most likely due to the increasing pet ownership practices in

the country (*Li et al., 2021*). Even so, they mentioned that pet allergies are still less common in China than in European nations, such as in Sweden (*Zhao et al., 2006*; *Lødrup Carlsen et al., 2012*; *Li et al., 2021*). It is argued that the area in which children live in plays a major role, with pet ownership in rural areas potentially serving to prevent allergies from developing, whereas in urban areas it may exacerbate them (*Krzych-Fałta et al., 2018*). Part of the reason might be that pet owners in urban areas are more prone to allow their companion in the house and their bedroom, which is likely to be due to living space limitations (*e.g.*, lack of outside yards) (*Krzych-Fałta et al., 2018*).

Based on the current evidence, the debate on the usefulness of pet avoidance offers contradictory arguments (*Chen et al., 2010*), hence no clear recommendation can be given. Keeping or not keeping a dog in the family should be decided based on other factors than the concern of developing asthma or allergies or of getting infected by a disease. It is essential to establish with the help of professionals (*e.g.*, veterinarians, behaviorists) efficient communication to help estimate the risk of zoonotic diseases as well as educate dog owners and non-owners (*Lipton et al., 2008*; *Speare et al., 2015*; *Overgaauw et al., 2020*). Including rabies prevention in educational curriculum for example has been shown to improve children's knowledge regarding the disease (*e.g.*, Malawi: *Burdon Bailey et al., 2018*; Philippines: *Amparo et al., 2019*). It took the form of one or several lessons on the subject, with or without a specially developed manual, introducing children to the animals that can transmit the disease, the symptoms and prevention, as well as safety around dogs and responsible pet ownership. Compared to children who had not received these lessons, those who had displayed better knowledge about both canine rabies and bite prevention up to 1 year after the intervention (*Burdon Bailey et al., 2018*; *Amparo et al., 2019*). However, greater knowledge is not always linked with a decrease in dog bites, highlighting the fact that the relationship between knowledge acquisition and human behavior change is complex and necessitates further investigation (*Amparo et al., 2019*).

## Fear of dogs (cynophobia)

The fear of dogs, also called cynophobia, is the experience of an irrational and persistent fear when exposed to a domestic dog. It can be a distressing problem for children which can interfere with their normal routine as well as the play and recreational activities of children and their families. Adults can also suffer from cynophobia but they generally report that their fear arose during childhood (*Doogan & Thomas, 1992*), emphasizing the importance of understanding the role of dogs in the lives of children. Phobias can be complex, involving genetic, maturational, and environmental factors (*King, Clowes-Hollins & Ollendick, 1997*). Parents often report that their child's fear arose after a dog attack or because the parents were themselves afraid of dogs (*King, Clowes-Hollins & Ollendick, 1997*; *May et al., 2013*). Beliefs play a significant role in the maintenance of phobias, stating that catastrophic predictions regarding a feared stimulus maintain phobic anxiety and that subsequent avoidance prevents disconfirmation (*Byrne et al., 2016*). The difference between children and adults is that a child with dog phobia may genuinely believe that a dog will attack them if they were to pet it whereas an adult will be aware that this outcome is unlikely and yet still experience high anxiety. Cynophobic children hold overestimated

beliefs regarding harm and that they were most concerned about dogs jumping on them (*Byrne et al., 2016*).

Methods to help cynophobic persons overcome their fear of dogs include exposure techniques which consist of exposing a phobic person to the stimulus that causes them fear in a safe environment. Some favor an exposure approach with actual dogs (*May et al., 2013*; *Tyner et al., 2016*; *Farrell, Kershaw & Ollendick, 2018*) while others take advantage of the advancement in technology to develop virtual reality applications (*Hnoohom & Nateeraitaiwa, 2017*; *Farrell et al., 2021*). Both methods have shown promising results with significant decreases in fear and sometimes "recovery". However, the control of the dog's behavior is usually a limiting factor for these techniques (*Calvo et al., 2013*).

Seven criteria exist in the diagnosis of a specific phobia, among which an anxiety response (*e.g.*, panic attack) and intense distress when exposed to the phobic stimulus (*American Psychiatric Association, 2013*). Knowing this, it could be argued that exposure therapy is unethical, especially with children where the concept of consent is debatable (*Gola et al., 2016*). Here, the end may not justify the means. Hence, other ways to treat cynophobia instead of evoking distress in a person who may be unwilling to engage in the therapy in the first place should be explored. The use of bibliotherapy (*i.e.*, using print materials to provide instructions normally provided by a therapist) for seven children has recently been explored and yielded promising results with significant reductions in fear severity and avoidance behavior as well as displays of good treatment adherence and retention (*Radtke et al., 2022*). Bibliotherapy and similar methods not relying on direct exposure to the source of fear should be further explored in future studies. In societies where it is possible to stumble upon a dog at any time, cynophobia can be a crippling condition and an important disturbance in positive child-dog interactions.

## Animal companion bereavement

Society does not always acknowledge the significance of pet bereavement, which can result in unresolved or unrecognized grief. Companion animals can sometimes be perceived as more dispensable when compared to humans (*Redmalm, 2015*), which explains why societal norms can deny the appropriate expression of grief following the death of a pet (*Kaufman & Kaufman, 2006*). While animal companions can help to make the human-loss mourning process less painful for children and adults through provision of their social support (*Kaufman & Kaufman, 2006*), there comes a time when it is the pet itself who dies. For dog owners, there are often no significant differences between the levels of grief severity experienced after the death of a human and a companion animal (*Lavorgna & Hutton, 2019*). As a matter of fact, the symptoms and characteristics associated with dog loss can be consistent with those associated with the death of a significant human, such as a close friend or family member (*Packman, Carmack & Ronen, 2011*).

Child grief is not expressed in the same manner as adults and is related to the child's developmental state (*Kaufman & Kaufman, 2006*), and this is consistent in their expression of pet bereavement (*Jarolmen, 1998*). The loss of a companion animal during childhood is no less important than the loss of a family member. It can be a life-changing event, especially for children for whom it may be their first significant loss with a profound

grief response (*Kaufman & Kaufman, 2006*). Not being appropriately supported during this hard time may lead to the development of complicated grief (*Kirwin & Hamrin, 2005*). Child and adolescent bereavement can result in depression, anxiety, social withdrawal, and behavioral disturbances (*Christ, Siegel & Christ, 2002*; *Kirwin & Hamrin, 2005*; *Kaufman & Kaufman, 2006*). In addition, the severity and prevalence of grief symptoms can be gender specific, with women reported to experience higher depersonalization (*i.e.*, feeling disconnected or detached from one's self) and death anxiety (*McCutcheon & Fleming, 2002*). Apart from gender, other variables influencing grief severity include closeness to the animal, perceived social support, and the type of death experienced by the animal (*McCutcheon & Fleming, 2002*; *Lavorgna & Hutton, 2019*). Those variables are generally linked: children are commonly those who rely the most on their pets for social support and who show more anger once the animal passes away (*McCutcheon & Fleming, 2002*). This anger may also be explained by the fact that, because of their young age, children do not consider the possibility of death and therefore have more trouble with understanding and accepting the situation when it arises (*Kaufman & Kaufman, 2006*). Despite the harshness of this experience, it can also teach children about the natural life cycle, which always includes death at some point (*Russell, 2017*; *Bowman, 2018*). It is important to appreciate the role that pets, and especially dogs, can have in children's lives in order to not trivialize the child's bereavement for their deceased canine friend (*Kaufman & Kaufman, 2006*).

## EFFECTS OF CHILD-DOG INTERACTIONS FOR DOGS

### Benefits

As a minimum level of care, dogs who live with people (also often described as '*owned*') are generally provided with food, shelter, and veterinary treatments. Children may take part in caregiving behaviors toward dogs (*Hall et al., 2016*; *Kerry-Moran & Barker, 2018*). Apart from taking care of their basic physiological needs, direct interactions with humans including children may offer benefits to dogs. When living closely with humans, dogs are able to establish attachment bonds with people which in turn may modulate their behavioral and emotional responses (*Nagasawa, Mogi & Kikusui, 2009*; *Merola, Prato-Previde & Marshall-Pescini, 2012*; *Wanser et al., 2020*). Interestingly, owner-dog dyads can present matching personalities. Indeed, using questionnaires (Big Five Inventory for humans and for dogs) completed by the owner and an independent peer person, *Turcsán et al. (2012)* found that all five personality dimensions examined (*i.e.*, neuroticism, extraversion, conscientiousness, agreeableness, and openness) showed significant positive correlations between adult owners and their dogs. This could be due to the "similarity-attraction hypothesis" which suggests that higher similarity between individuals lead to higher attraction between them (*Byrne, Griffitt & Stefaniak, 1967*). To date, no similar research on child-dog dyads has been undertaken.

Dogs can synchronize their behavior with that of children from their family (*Wanser, MacDonald & Udell, 2021*). During walking sessions, dogs exhibited activity, proximity, and orientation synchronization with the child who was walking with them at higher rates than would be expected by chance (*Wanser, MacDonald & Udell, 2021*). Although at lower rates than when walked by their adult caregivers (*Duranton, Bedossa & Gaunet, 2018*,

*2019*), those results demonstrate that dogs may perceive familiar children as social partners. Additionally, children provide dogs with a source of social companionship and create opportunities for recreational activities (*Hall, Finka & Mills, 2019*). Finally, petting (*e.g.*, tactile stroking, patting) has been shown to have marked effects upon the autonomic functioning of dogs. Indeed, while being petted, dog heart rate decreases which may relate to reduction of stress as a result of being touched (*Csoltova et al., 2017*; *Mariti et al., 2018*).

In the same way that dog walking can be beneficial for people, it is equally advantageous for dogs. Walking dogs has been identified as very important for dog wellbeing. For households that do not include a yard, walks enable the dogs to relieve themselves outside. In addition, walks beyond the house or property boundaries offer a perfect opportunity for the dog to exercise. This physical activity can help prevent dog obesity (*Bland et al., 2009*), while also providing mental stimulation (*American Veterinary Medical Association, 2022*) and opportunities to sniff in known and new environments (*Kokocińska-Kusiak et al., 2021*). Using a cognitive bias test, *Duranton & Horowitz (2019)* showed that the practice of nosework, or an olfaction-based activity, with their owners increased optimism in dogs. They argued that when dogs practice nosework, they can express their natural behavior, a key point for positive animal welfare (*Mellor, 2016*). Walks can also offer the opportunity for dogs to socialize with conspecifics, either en route, or at a destination such as a dog park (*Westgarth et al., 2010*). Proper walking practices (*e.g.*, allowing dogs to sniff their environment, giving them time to socialize) should be taught to dog owners and their families to favor positive dogs' experience when being walked and thus enhance their welfare.

## Risks

While a lot of attention has been given to the effects of children and dogs interactions for people, little attention has been paid to the risk of human interactions to dog quality of life (*Hall, Finka & Mills, 2019*). The complex nature of the environment in which dogs live may place them in recurrent or chronic states of stress which can have long-term outcomes for the dog quality of life. Children may be part of this environment and because of their unpredictable and active mannerisms as well as their difficulty in identifying subtleties of behavior (*Meints, Brelsford & De Keuster, 2018*), they may put their canine companion under distress, possibly increasing the risk of aggression toward children. However, euthanasia or relinquishment are often the consequences for dogs showing aggression towards people (*Casey et al., 2014*). It is thus crucial to pinpoint the specific factors in interacting with children which may represent a threat to dog wellbeing and general quality of life.

Some risk factors which can distress dogs are spatial restriction, social isolation, changes in routine, loud noises, and unexpected events (*Hall, Finka & Mills, 2019*). A number of child-dog interactions may jeopardize dog quality of life: "unprovoked child attention" (*e.g.*, rough contact), "interaction and environmental unpredictability" (*e.g.*, meltdowns and tantrums, need for appropriate recreational activities), and "child games" (*e.g.*, playing "dressup", loud games) (*Hall, Finka & Mills, 2019*). Furthermore, while some dogs can respond favorably to being petted, it is important to consider that some common physical

interactions may be perceived as unpleasant by certain dogs: some individuals dislike being touched on the top of the head or being hugged for example (*Kuhne, Hößler & Struwe, 2014*), others may have injuries or past experiences with people which make them unwilling to be touched. Although focusing on human-cat interactions, a set of guidelines aiming to enhance companion cats' comfort when interacting with humans was recently created (*Haywood et al., 2021*). When people followed those guidelines, the frequency and duration of affiliative and positively-valenced behaviors in shelter cats were significantly greater, and human-directed aggression decreased (*Haywood et al., 2021*). Future studies assessing similar guidelines in the context of human-dog interactions, and especially child-dog interactions in our case, should be undertaken. The principle of "consent test" appears promising: when petting a dog (or any other animal), take a pause to see what they do, then respond accordingly will give the opportunity for the animal to choose when and for how long they are being petted (*Todd, 2022*). Whereas the above examples depict interactions during which children do not intentionally intend to harm their companions, childhood acts of animal cruelty also exist (*McDonald et al., 2018*; *Hawkins, Scottish Society for the Prevention of Cruelty to Animals & Williams, 2020*).

Despite having co-evolved with people, dogs do not choose their modern-day owners and the people with whom they will interact throughout their lifetime. Indeed, dogs and especially companion dogs are living in a "human's world" where we are the one largely defining and managing almost every aspects of their lives (*Benz-Schwarzburg, Monsó & Huber, 2020*). Some dogs will not get along with children in a way that is perceived positively. Dogs who have been in the family for longer than the child exhibit less affiliative behaviors toward them (*Arhant, Beetz & Troxler, 2017*). In the same study, parents of children aged 6 months to 3 years were the ones reporting the highest levels of child avoidance in their family dog (*Arhant, Beetz & Troxler, 2017*). While introducing dogs to children during their socialization period can enhance better behavior (*Arai, Ohtani & Ohta, 2011*), being obligated to engage in non-optimal relationships can increase dogs' chronic stress levels and consequently diminish their welfare (*Cimarelli et al., 2019*).

Although dog walking can be very important for canines, a rather large proportion of dog owners do not walk their dogs (*e.g.*, Japan: 35%, *Oka & Shibata, 2012*; USA: 30%, *Coleman et al., 2008*). Understandably, children are rarely allowed to walk the family dog on their own, rather they join their parents in walking activity (*Salmon et al., 2010*). Thus, if adults do not walk their dogs, there are few chances that the dog will be walked at all. Apart from a lack of physical activity, other factors dog- and/or owner-dependent can lead to the onset of obesity in dogs. Owner-dependent factors include food type and feeding rate (*Mao et al., 2013*), and children can sometimes overfeed their canine companions with treats resulting in weight gain. Unfortunately, the incidence of canine obesity is ever increasing (*German, 2006*; *German et al., 2018*). Cross-sectional studies in the UK, Spain, and China reported 65%, 41%, and 44% of overweight dogs respectively (*Mao et al., 2013*; *Montoya-Alonso et al., 2017*; *German et al., 2018*). However, retrospective studies yield less alarming results. In the UK, *Pegram et al. (2021)* have estimated that 7% of dogs under veterinary care in 2016 were overweight using electronic patient records. Using the same method, results from New Zealand reported 28% of overweight dogs (*Gates et al., 2019*).

Veterinary clinical records may underreport overweight status in dogs, as this discrepancy of results between methodologies appears to highlight (*Rolph, Noble & German, 2014*). Canine obesity can be associated with numerous health issues such as osteoarthritis, cardiovascular disease, diabetes mellitus, and others, all of which can significantly reduce the quality of life as well as the lifespan of the dog (*Laflamme, 2012*; *Endenburg et al., 2018*). Additionally, children (and adults) are sometimes prone to giving food items to dogs that can turn out to be harmful to them, such as chocolate. This can, in some extreme cases, lead to the death of the animal (*Weingart, Hartmann & Kohn, 2021*).

It is important to consider the welfare of working dogs, such as therapy and assistance dogs, that are involved in numerous settings around the world (*Cobb, Otto & Fine, 2021*). While therapy dogs take part in structured, therapeutic interventions accompanied by licensed professionals (*Schoenfeld-Tacher et al., 2017*), assistance dogs (also called service dogs) permanently live with the humans whose daily life they are meant to assist (*Winkle, Crowe & Hendrix, 2012*). In contrast to the rather large literature on the effects of animal-assisted interventions on humans, few studies assessed their impact on dogs (*Glenk, 2017*). Research has used different physiological and behavioral measures as well as handler surveys to assess stress in this population (*Burrows, Adams & Millman, 2008*; *Marinelli et al., 2009*; *Palestrini et al., 2017*; *McCullough et al., 2018*; *Uccheddu et al., 2019*; *Melco et al., 2020*). Although clear conclusions cannot yet be drawn about the impact of animal-assisted interventions on dog wellbeing (*Glenk, 2017*; *Glenk & Foltin, 2021*), there are records of inappropriate behaviors and mistreatment which can lead to the deterioration of dogs' health (*Heimlich, 2001*; *Hatch, 2007*). Inappropriate behaviors may come from the handlers or the intervention recipients. Even a highly trained dog can still be scared of certain objects, and a handler forcing it close to those objects can reinforce the fear and create anxiety (*Hatch, 2007*). Moreover, it happens that handlers refrain from providing water to their dogs for various amounts of time for fear of the dog urinating in inappropriate areas (*Hatch, 2007*). By contrast, some children involved in animal-assisted interventions can exhibit aggression toward the animals and therefore should be kept away to prevent incidents (*Heimlich, 2001*).

The welfare of free-roaming dogs is generally perceived and described as very poor (*Jackman & Rowan, 2007*; *Cobb, Lill & Bennett, 2020*). Although not directly linked to child-dog interactions, those dogs commonly suffer from malnutrition, dehydration, and diseases (*Matter & Daniels, 2000*). These animals receive little veterinary care and thus present high rates of mortality (*Jackman & Rowan, 2007*). Because some are carriers of rabies and attack humans, especially children, and despite the vaccination effort in many countries, some take matters into their own hands and kill the canines (*Cleaveland et al., 2006*; *Jackman & Rowan, 2007*). However, in some respects, these dogs live their lives with far greater agency in terms of social and environmental choices than dogs living in human homes (*Cobb, Lill & Bennett, 2020*), and could be considered to enjoy better welfare in those aspects than dogs living in close contact with people.

## FUTURE DIRECTIONS

Future research should seek to better understand the role of attachment between dogs and children in relation to the physical and mental health benefits of dog ownership (*Purewal et al., 2017*). Several tools to measure the dog–human relationship exist and have been reviewed by *Payne, Bennett & McGreevy (2015)*. While the attachment of children towards dogs has been the focus of a number of studies (*e.g.*, *Marsa-Sambola et al., 2016*; *Wanser et al., 2019*), research on the attachment of dogs towards children is still scarce.

The attachment subscale of the Canine Behavioral Assessment & Research Questionnaire (C-BARQ; *Serpell, 2022*) could be an interesting area of investigation. Apart from bi-lateral attachment, studying the possibility of personality matching between children and dogs could also provide new insights into successful relationships.

The role of culture in attitudes towards animals and pet-keeping practices should be further investigated (*Jackman & Rowan, 2007*; *Gray & Young, 2011*; *Jegatheesan, 2015*). Because some subjects have only been studied in a few countries (Fig. 3), it is not possible to make broad generalizations from these results. In countries where dog-borne zoonoses such as rabies are still prevalent, people's perceptions of dogs are likely to be related to these risks (*Tiwari et al., 2019*). To the authors' knowledge, no study comparing the prevalence of cynophobia between countries exists. Yet, in countries still experiencing deadly diseases such as rabies, it seems fair to suggest that cynophobia could have an evolutionary purpose. Indeed, it could increase survival as being afraid of dogs is likely to reduce the risk of getting bitten due to avoidance of the animals. Thus, future research could try to determine the moderating role of culture on child-dog interactions, for example by comparing children's attitudes towards animals between countries. We also believe it to be a good opportunity to study interactions between children and stray/free-roaming dogs, seeing as such populations are not present in every country over the world and could therefore bring new insights into the diversity of child-dog interactions around the world.

Despite a growing interest in the use of robot dogs (*e.g.*, Sony's robotic dog AIBO), there is still a lot to discover regarding its impacts on children. To date, research has shown that children can display interest in robot dogs and can be seen interacting with them in the same way they would with a live dog (*Melson et al., 2005*; *Ribi, Yokoyama & Turner, 2008*; *Weiss, Wurhofer & Tscheligi, 2009*). Moreover, it appears that child-robot interactions can present similar benefits to children as child-dog interactions. Namely, interacting with a dog robot can help in the social development of children (*e.g.*, neurotypical children: *Heljakka, Lamminen & Ihamaki, 2021*; children on the autistic spectrum: *Stanton et al., 2008*). Robot dogs can represent a very good opportunity to provide the benefits of child-dog interactions to children allergic to dogs. Additionally, the use of robot dogs could also be investigated for treating cynophobia.

Human-animal interactions research most often concentrates on the impacts on humans. Consequently, there is a gap in the understanding of the impact of children on dogs and canine welfare (*Hall, Finka & Mills, 2019*). To give just one example, our review presents information on the impact of child-dog interactions on children and adolescents.

Future research should seek to do the same with dogs, and analyze the impact of such interactions on dogs at different ages. Two themes may explain this dearth of knowledge, namely scientific communication and research funding (*MacLean et al., 2021*). As shown by the increase almost every year in the number of papers related to those subjects (Fig. 1), dog popularity fuels public interest in canine science and the impact of dogs on human health and wellbeing. However, research in child-dog interactions often offers contradictory results. This is most likely due to the wide diversity in methodologies, small effect sizes, and homogenous samples (*Purewal et al., 2017*). Cross-sectional and correlational study designs do not enable causal inferences to be clearly made. Yet, good scientific communication requires honesty, relevance and effectiveness (*MacLean et al., 2021*). Although we realize the great amount of time and effort that it would necessitate, longitudinal studies would be better suited to understand the impact of child-dog interactions. Science continues to explore and identify situations that may undermine dog welfare, and safeguarding their wellbeing remains a crucial area of study, even more so in the context of animal-assisted interventions (*Glenk & Foltin, 2021*).

## CONCLUSION

In summary, growing up alongside one or several dogs has become a common occurrence in the life of numerous children around the world (*Melson & Fine, 2015*). Knowing the various outcomes possibly derived from child-dog interactions can help weigh the pros and cons of living alongside a companion dog. Current evidence suggests that child-dog interactions may be beneficial to both. All the different aspects of wellbeing, *i.e.*, physical, mental, and social, are interconnected, meaning that improving one's physical wellbeing may lead to an improvement of their mental and social wellbeing as well, creating a virtuous circle. Generalizing the findings from existing studies should be done with caution. Yet, in spite of the use of various methodologies that can lead to weaknesses (*Purewal et al., 2017*), results show that dogs have the potential to improve children lives just as children contribute to the quality of life of dogs. Nonetheless, such interactions can also bring about negative outcomes such as bite injuries, dog-borne zoonoses, or stress. Overall, the benefits of child-dog interactions seem to outweigh the risks for children but not dogs. However, the evidence suggests that by supervising those encounters and by increasing people's knowledge about dog behavior and the possible outcomes of child-dog interactions, we could increase the positive effects all the while reducing the negative ones that are, for the most part, preventable. The mechanisms through which both species can promote each other's wellbeing require further investigation. There is little knowledge so far on the potentially differential effects of culture on physical, mental, and social outcomes. Lastly, longitudinal and controlled designs that allow repeatability should be favored in future studies. Further research investment to optimizing child and dog interactions will underpin the success and sustainability of our long-term relationship with dogs as companions.

## ACKNOWLEDGEMENTS

We thank Dr. Evan MacLean for his valuable comments on an early version of this review. We also thank the editor, Dr. Savino, and the reviewers, including Dr. Volsche, and two anonymous reviewers, for their helpful advice. We would also like to thank some present and past dogs, Brennus, Hetian, Jack, Luna, Qiemo, Rudy, and Tucker, who represent an endless source of motivation to study the human-dog relationship.

### Funding
The authors received no funding for this work.

### Competing Interests
Alan G. McElligott is an Academic Editor for PeerJ.

### Author Contributions
- Claire S. E. Giraudet conceived and designed the experiments, performed the experiments, analyzed the data, prepared figures and/or tables, authored or reviewed drafts of the article, and approved the final draft.
- Kai Liu conceived and designed the experiments, authored or reviewed drafts of the article, and approved the final draft.
- Alan G. McElligott conceived and designed the experiments, authored or reviewed drafts of the article, and approved the final draft.
- Mia Cobb conceived and designed the experiments, prepared figures and/or tables, authored or reviewed drafts of the article, and approved the final draft.

### Data Availability
    This is a scoping review and does not have raw data.

### Supplemental Information
Supplemental information for this article can be found online at http://dx.doi.org/10.7717/peerj.14532#supplemental-information.

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
