# Peer review of "Are children and dogs best friends? A scoping review to explore the positive and negative effects of child-dog interactions"

_PeerJ, doi:10.7717/peerj.14532_

## Round 0.1 · original submission · Minor Revisions

This article is interesting, but needs a revision following the suggestion of the three reviewers.

Further, I suggest Authors improve the abstract by inserting more data i.e. temporal period covered and more details of the results.

Reviewer 1 ·

Basic reporting

The paper is clearly written. English is correct and professional.
The references are appropriate.
The article structure can be improved by adding informative tables that provide some detail about the type of article, quality, riks of biases, etc., as usually done in systematic reviews (including scoping reviews).

The paper addresses and interesting topic that has interdisciplicary relevance.

The Introduction is generally adequate, even though it could be more focused on the specific aims of the review without too many sentences on the history of dog domestication (unless this is linked to the questions of the review).

Experimental design

The Methods section needs more work in specifying which types of articles were retrieved: observational studies, reviews, commentaries, case reports? How large were these studies? The reader must have some idea of the source of the information provided in the review. For example, lines 345-346 state that "Taking care of a dog, taking it for walks, feeding it, and playing with it can promote children certain social values and skills, for example a sense of responsibility for the welfare of others (Muldoon, Williams & Lawrence, 2016).". Is this the personal opinion of Muldoon et al., or is it derived from data collected as part of some sort of systematic study?

Validity of the findings

An analysis of the quality of the reports would help determine the validity of the statements included in the review. The statements in the Results section have references but it is unclear if that reference is of a study with data or just personal opinion or impression.

It would help to add one or two tables to summarize the types of articles and, if data driven, the sample size and study design, and to report the main findings.

·

Basic reporting

The manuscript, "Are children and dogs best friends?" is well written, thoroughly researched, and properly structured. This review is novel in its breadth, as well as the effort to take into consideration the impact of these relationships on the children and the dogs. This attention to the nonhuman is rare in human-animal interaction research, especially when children are involved. The authors do an excellent job of emphasizing this point in the abstract and in lines 112-113. This attention to both species makes this a novel review.

The introduction is clear, easy to follow, and sets up the importance of studying child-dog interactions. It provides sufficient information to prepare the reader for the content of the review without being unnecessarily long.

Experimental design

This manuscript is well suited to PeerJ. As a literature review, it is thorough in the breadth and depth of articles included. The investigation included numerous articles of importance to the topic, and the qualitative approach - as opposed to a meta-analysis - is refreshing. To emphasize the extensive inclusion of related articles, the References list is nearly as long as the article itself.

The organization of the review is logical, and the categories (impacts on children, impacts on dogs; beneficial vs risks) makes it easy to follow. I was pleased to realize there were citations I had not heard of, and found myself making note of articles to seek out at a later date.

Validity of the findings

The validity of the authors' points are clearly supported by the extensive literature that has been synthesized into this article. There is a note suggesting more work needs to be done regarding the impact of attachment on these relationships. I fully agree. In addition to child to dog attachment, perhaps the attachment subscale of the C-BARQ could be used to investigate dog to child attachment, too.

This is a well timed, and I think, much needed review. I suspect it will be heavily cited within the human-dog interactions literature.

Additional comments

I am particularly appreciative of the cross-cultural approach to this paper. Granted, it is still hindered by the inclusion of English manuscripts only (there are some great papers in Portuguese and Spanish), but that is a by-product of language knowledge of the team. Regardless, it was refreshing to see data from the US, UK, Australia, Japan, China, Iran, India ... the list goes on, and this broader perspective is definitely needed in reviews. This is especially true in HAI or AAI papers.

Reviewer 3 ·

Basic reporting

The review is within the scope of the journal. There are other reviews about child-dog interactions that focus on one topic (please cite them in L116), but this review has a broader scope and is of relevance to a wider audience, so there is a valid reason for it, and it is great that the search found such a huge range of studies.

The introduction starts with a long description of dog domestication, which I do not think is relevant to introduce the subject of the review (child-dog interactions). I suggest only keeping the first sentence of the first paragraph and continuing from the second paragraph (i.e., delete L61-79). Otherwise, the introduction adequately introduces the subject and the goals are clear.

In general, there are quite a lot of English language issues that need fixed (mostly basic things like missing articles and missing commas).

Experimental design

L140: The keywords used to conduct the search seem to be biased towards the effect of dogs towards children (e.g., ‘education’ and ‘bite’). They could have searched keywords related to dogs, such as ‘welfare’. Also, ‘bite’ has negative connotations, but they did not include a positive word, such as ‘play’, so the search may be biased.

Sources are adequately cited but require a bit more information to understand what the studies found (e.g., L189, L245 and L291). For example, I do not understand how therapy-dog walking can help if a child has orthopaedic limitations? (L252)

The review could be better structured and organised in more coherent subsections.
The first aim: the first paragraph in ‘How children and dogs interact with other’ (L166) should be moved to the Introduction, as it does not describe child-dog interactions and provides a good background about what situations children may interact with dogs.
This section should be purely descriptive but L192-198 already describes the negative effects of child-dog interactions for dogs, so this should be moved to the appropriate section. Furthermore, there could be subsections within this section, e.g., home activities, education, animal-assisted interventions, etc.

The second aim: they start with subsections on the physical, mental and social benefits for children, but then the subsections on the risks for children are categorised differently, so it would be better if they also followed the same pattern of physical, mental and social risks for children. Furthermore, they do not break down the benefits and risks into physical, mental and social domains for dogs.

Validity of the findings

The abstract and introduction states that they will review the effects of child-dog interactions on the physical, mental and social wellbeing of both species, however the review is unbalanced and focuses much more on the effects of child-dog interactions for children than dogs. This could be because the keyword search was biased towards children or there are fewer studies on the effects for dogs, but this should be addressed because it currently does not meet the goals set out in the Introduction.

The first paragraph of ‘Future directions’ (L829) does not address the question about child-dog interactions specifically but rather more generally about the role of culture in attitudes towards animals, so it needs to be stated more clearly (e.g., comparing children’s attitudes towards animals between countries). However, the second paragraph (L842) does adequately identify the gap in the understanding of the impact of children on dogs and their welfare. Finally, the authors do not have a clear position in the Conclusion – as a reader, it seems like there are not many positive effects of child-dog interactions for dogs based on their findings.

Additional comments

The abstract could be more concise, e.g., remove sentences like “walking a dog and playing with one are perfect physical activities”. The abstract lacks a sentence on the overall conclusion of the review (it seems the benefits of child-dog interactions seem to outweigh the risks for children but not for dogs). It would also be good to mention who the audience is (i.e., it is relevant for paediatricians, psychologists, veterinarians, dog owners etc.).

The paragraph about stray dogs (L222) and the welfare of free-roaming dogs (L817) seem out of place and not related to child-dog interactions, so I suggest completely removing free-roaming/stray dogs from the review, as there is barely any research on it. Instead, these could be emphasised in the ‘Future directions’ section.

Overall, I think the authors could relate the second part of the review (positives and negatives of child-dog interactions) back to the first part (describing different interactions) more. For example, in the first part, they mention co-sleeping with pets, but they do not follow this up in the second part – is there any information about whether this has a positive or negative effect on children? If there are no studies, this could be mentioned in ‘Future directions’.

There are some sentences that do not fit into the sections (e.g., L397 about the peer-review process, and L867) and are not relevant to this review (e.g., L700 about dog-owner personalities, as this is about correlations between adult owners and their dogs).

There are several positives and negatives effects of child-dog interactions for children and dogs that I thought would be important to discuss but were not mentioned – this could be because the search did not find any studies about these topics, but they could be mentioned in ‘Future directions’. For example:
• A potential physical risk of dog-walking for children is that it can be dangerous if a dog is not well-trained to walk on a lead or if they are very strong, they might pull the child over or into the road.
• Dog bites (physical risk for children): there is no mention that dog bites may happen if they play too rough with a dog or make the dog feel uncomfortable (but they do mention it in L749 in the risk for dogs’ section).
• L630: as there is a paragraph of methods to help cynophobic persons, I think you could also discuss methods to help children who are allergic to dogs and cannot interact with them and would miss out on the benefits of child-dog interactions (e.g., increase positive social behaviours and empathy) or would benefit from animal-assisted therapy (e.g., going through potentially stressful tasks, such as medical examinations). Perhaps the use of robotic dogs could help them in these cases – this could be included when you discuss the advancement of technology to help with cynophobia in L634. Furthermore, to use of robotic dogs has been shown to benefit children with autism (see references at the end). Perhaps this could be mentioned in the ‘Future directions’ section.
• This review shows the different effects of child-dog interactions for children and adolescents, and this may also be the case for dogs of different ages. For example, there may be more positive effects for puppies who grow up with children and become socialised and desensitised to screaming children, whereas this scenario may stress out an older dog who is not used to interacting with children.
• A potential physical risk for dogs is that children may feed them something harmful like chocolate, even if the child has good intentions.
• L791-802: The issue of canine obesity is important to discuss as a risk for dogs, but the information is not relevant to child-dog interactions – if the dog is not walked, it is not usually the child’s responsibility. Therefore, it would be better to discuss canine obesity in terms of children potentially overfeeding the dog with treats or leftovers.

L153 Synthesis of results: I do not think you need to keep referring to Table 1 three times in the same paragraph.

L408: This is a long list of references, consider presenting it in a table to make it easier to read which country and the reference for it.

L449: I don’t think separation solves the problem of biting children, it only prevents it, which might not be the best advice for progression in child-dog relationships, and this may be detrimental to the dog’s wellbeing.

L464: The example of baby photos with dogs is not a bad interaction in itself, so I think this could be worded differently and is nicely backed up by the next sentence – safe cohabitation (and safe interactions) is based on mutual understanding of interspecific signalling and social gestures.

L492: It is possible that rather than the increased time with a dog from the “stay-at-home” order, an increased number of people who acquired a dog during the pandemic and did not train/socialise their dog well (maybe because they couldn’t take their dog out to interact with other people and dogs) could have led to more child injuries. This section could be phrased more carefully.

L580: Why not cite all eight allergens? It might be useful for readers who are searching for a specific allergen and I do not think it is too much to list them all.

L803: What kind of animal-assisted interventions were in these papers? If they are specific to children, please state it. If not, I do not think this information is relevant for this review.

References for papers on children-robot interactions:
https://doi.org/10.1145/1349822.1349858
https://doi.org/10.1145/1056808.1056988
https://doi.org/10.2752/175303708X332053
https://doi.org/10.1145/985921.986087

---

## Round 0.2 · accepted · Accept

Now the article is suitable.

Reviewer 1 ·

Basic reporting

The revised version of this article is responsive to the previous critique.
This review represents an informative report on a important theme and can be a good contribution to the literature.

Experimental design

Standard methods have been applied. Appropriate to the study.

Validity of the findings

Good validity of the results.

·

Basic reporting

The authors have addressed my comments, and I am in support of publication at this time.

Experimental design

The authors have addressed my comments, and I am in support of publication at this time.

Validity of the findings

The authors have addressed my comments, and I am in support of publication at this time.

Additional comments

The authors have addressed my comments, and I am in support of publication at this time.

Reviewer 3 ·

Basic reporting

-

Experimental design

-

Validity of the findings

-

Additional comments

-